# Herpes simplex virus 1 regulates β-catenin expression in TG neurons during the latency-reactivation cycle

**Kelly S. Harrison**[1], **Liqian Zhu**[1,2¤], **Prasanth Thunuguntla**[1], **Clinton Jones**[1]*

**1** Department of Veterinary Pathobiology, College of Veterinary Medicine, Oklahoma State University, Stillwater, OK, United States of America, **2** College of Veterinary Medicine and Jiangsu Co-innovation Center for Prevention and Control of Important Animal Infectious Diseases and Zoonosis, Yangzhou University, Yangzhou, China

¤ Current address: Department of Ophthalmology and Visual Sciences, University of Wisconsin, Madison, WI, United States of America

* clint.jones10@okstate.edu

**Data Availability Statement:** All relevant data are within the paper.

**Funding:** This research was supported by grants from the National Institute of Neurological

## Abstract

When herpes simplex virus 1 (HSV-1) infection is initiated in the ocular, nasal, or oral cavity, sensory neurons within trigeminal ganglia (TG) become infected. Following a burst of viral transcription in TG neurons, lytic cycle viral genes are suppressed and latency is established. The latency-associated transcript (LAT) is the only viral gene abundantly expressed during latency, and LAT expression is important for the latency-reactivation cycle. Reactivation from latency is required for virus transmission and recurrent disease, including encephalitis. The Wnt/β-catenin signaling pathway is differentially expressed in TG during the bovine herpesvirus 1 latency-reactivation cycle. Hence, we hypothesized HSV-1 regulates the Wnt/β-catenin pathway and promotes maintenance of latency because this pathway enhances neuronal survival and axonal repair. New studies revealed β-catenin was expressed in significantly more TG neurons during latency compared to TG from uninfected mice or mice latently infected with a LAT$^{-/-}$ mutant virus. When TG explants were incubated with media containing dexamethasone to stimulate reactivation, significantly fewer β-catenin+ TG neurons were detected. Conversely, TG explants from uninfected mice or mice latently infected with a LAT$^{-/-}$ mutant increased the number of β-catenin+ TG neurons in the presence of DEX relative to samples not treated with DEX. Impairing Wnt signaling with small molecule antagonists reduced virus shedding during explant-induced reactivation. These studies suggested β-catenin was differentially expressed during the latency-reactivation cycle, in part due to LAT expression.

## Introduction

More than 50% of adults in the United States are latently infected with herpes simplex virus 1 (HSV-1), reviewed in [1–3]. HSV-1 infections at the surface of the eye or oral cavity lead to life-long latent infections within sensory neurons of trigeminal ganglia (TG) as well as neurons within the central nervous system [4, 5]. Sporadic virus reactivations occur during

Disorders and Stroke of the National Institutes of Health under Award Number R21NS102290 (CJ), USDA-NIFA Competitive Grants Program (16-09370 and 2018-06668) (CJ), the Sitlington Endowment (CJ), support from the Oklahoma Center for Respiratory and Infectious Diseases (National Institutes of Health Centers for Biomedical Research Excellence Grant # P20GM103648), and funds from the OSU CVM Research Advisory Council (KH). These studies were also partially supported by the Chinese National Science Foundation Grant (No. 31472172 and No. 31772743) (LZ) and National Key Research Program (No. 2016YFD0500704) (LZ).

**Competing interests:** The authors have declared that no competing interests exist.

the life of an infected individual resulting in virus shedding at the periphery, which can trigger recurrent disease.

In contrast to productive infection where more than 70 viral transcripts are readily detectable, including the latency-associated transcript (LAT), LAT is the only viral transcript abundantly expressed during latency [2, 3]. Deleting LAT coding sequences or LAT promoter sequences reduce the efficiency of reactivation in latently infected rabbits and mice [1–3]. This complex locus expresses multiple transcripts, six micro-RNAs, and two small non-coding RNAs [6–8]. LAT protects neurons from cell death, partly by inhibiting apoptosis and viral gene expression [9–13]. The anti-apoptosis functions of LAT are dependent on an active serine/threonine protein kinase (AKT) [14, 15]. LAT also interferes with granzyme B mediated apoptosis [16] and increases CD8+ T cell exhaustion during latency [17]. The anti-apoptosis functions of LAT are important in the context of the latency-reactivation cycle because inserting anti-apoptosis genes into the LAT locus of a LAT$^{-/-}$ mutant virus restores reactivation to WT like levels [18–21]. As LAT is generally considered to express non-coding RNAs, LAT may also influence expression of cellular factors that promote the establishment and maintenance of latency.

Cellular genes associated with the Wnt/β-catenin signaling pathway are differentially expressed during the bovine herpesvirus 1 (BoHV-1) latency-reactivation cycle [22–24]. For example, during latency signficantly more TG neurons express β-catenin relative to TG from uninfected calves or during reactivation from latency. When BoHV-1 reactivation from latency is induced by the synthetic corticostroid dexamethasone, the Wnt/β-catenin signaling pathway is turned off, in part because expression of several soluble Wnt antagonists are stimulated [37, 38]. The Wnt/β-catenin signaling pathway is comprised of 19 related Wnt receptors, 7 Wnt co-receptors, many downstream effectors of this pathway, and two distinct families of Wnt antagonists [37, 38]. In the absence of the Wnt ligand or in the presence of a Wnt antagonist, a multi-protein β-catenin destruction complex, which includes Axin, APC (adenomatous polyposis gene), glycogen synthase kinase 3 beta (GSK3β), and casein kinase alpha (CKIα), hyper-phosphorylates β-catenin. This leads to poly-ubiquitination of β-catenin and degradation via the proteasome. Binding of Wnt to one of its receptor/co-receptor complexes disrupts the β-catenin destruction complex, the transcription factor β-catenin levels increase, β-catenin enters the nucleus, and interacts with TCF (T cell factor) family members bound to DNA. β-catenin binding to a TCF family member displaces transcriptional co-repressors and recruits co-activators to activate Wnt target genes. This signaling pathway promotes neuronal development and maturation, axonal growth and targeting, and neuronal survival [25–30], functions crucial for maintaining latency. Since BoHV-1 is a neurotropic α-herpesvirinae subfamily member, it is reasonable to suggest the canonical Wnt/β-catenin signaling pathway may also regulate certain aspects of the latency-reactivation in additional neurotropic herpesviruses.

In this study, we provide evidence that significantly more TG neurons expressed the β-catenin protein in mice latently infected with HSV-1 compared to TG from uninfected mice or mice latently infected with a LAT$^{-/-}$ virus. Furthermore, β-catenin protein expression in TG neurons was reduced to levels similar to uninfected mice following TG explant for 4 or 8 hours if incubated with media containing the synthetic corticosteroid dexamethasone (DEX) to enhance reactivation. Virus shedding during explant-induced reactivation was significantly reduced relative to controls when TG from mice latently infected with WT HSV-1 were treated with a β-catenin specific inhibitor (iCRT14) or another Wnt antagonist (KYA1797K). Collectively, these studies suggest β-catenin expression in TG neurons may play a role during the latency-reactivation cycle.

## Materials and methods

### Virus and cell lines

HSV-1 strain McKrae dLAT2903R (WT; LAT$^{+/+}$) and dLAT2903 (LAT$^{-/-}$) were kindly gifted from Steven Wechsler and grown in Vero cell (ATCC CCL-81) monolayers in minimal essential medium (MEM) supplemented with 10% fetal bovine serum (FBS), 2mM L-glutamine, 100 I.U./mL Penicillin, 100μg/mL Streptomycin at 37˚C, 5% $CO_2$ until >80% CPE was observed. Viral aliquots were obtained through serial freeze-thawing of cells and subsequent centrifugation to remove cellular debris. Virus was titered on Vero cell monolayers to determine PFU/mL for each stock prior to animal infections.

### Animals

The mouse studies performed in this study were reviewed by the Oklahoma State University Institutional Animal Care and Use Committee. The mouse studies were approved and the animal use protocol is VM15-26. Female, 6–8 week-old Swiss-Webster mice were purchased from Charles River Laboratories. Mice were acclimated to standard laboratory conditions (12 hrs. light/dark cycles, 5 animals/cage) for 1 week prior to ocular infections. For infections, mice were anesthetized via standard isoflurane/oxygen vaporization and infected with ~2x10$^5$ PFU wt HSV-1 (n = 50) or the LAT$^{-/-}$ mutant (dLAT2903, n = 50) in 2μL MEM per eye without scarification as described previously [31]. Ocular infections can result in herpes simplex encephalitis (HSE) of approximately 50% of animals during the first 15 days after infection. Consequently, mice were monitored twice daily for the duration of the experiments and, if any HSE-related symptoms such tremors, swollen forehead, or weight loss were observed, animals were promptly and humanely euthanized via isoflurane overdose and subsequent cervical dislocation. Infected mice were not allowed to reach HSE-induced death as an endpoint. A subset of surviving animals commonly displayed symptoms of acute ocular herpes infections such as redness, discharge, and fur-loss at the site of inoculation and were therefore treated twice daily with Neomycin/polymyxin B/Bacitracin zinc triple antibiotic ophthalmic ointment (Bausch + Lomb, Tampa, FL) to prevent secondary bacterial infection and minimize pain and distress. Mice infected for at least 30 days were operationally defined as being latently infected.

### Immunohistochemistry

TG from uninfected (U, n = 5), LAT$^{-/-}$ (n = 25 mice, 5 mice/group: Latency; 4h; 4h +Dex; 8h; 8h+DEX) or WT latently infected mice (n = 25, 5 mice/group: Latency; 4h; 4h +Dex; 8h; 8h +DEX) were harvested at least 30 days after infection via humane euthanasia by isoflurane overdose followed by decapitation at the atlanto-occipital joint. TG were dissected out and either directly placed in formalin or explanted in MEM containing 2% stripped serum with or without DEX addition for 4 or 8 hours and then formalin fixed. Slide preparation and IHC was performed as previously described [31]. Briefly, 4–5 μm sections of paraffin-embedded TG were cut and mounted onto glass slides. Slides were incubated at 65˚C for ~20 mins before deparffinization in Xylene and serial rehydration using decreasing concentrations of ethanol. Endogenous peroxidases were blocked by incubating slides in 0.045% $H_2O_2$ for 20 minutes. Antigen retrieval was performed using Proteinase K (Agilent Dako, Santa Clara, CA) and slides were blocked using Animal Free Blocking Solution (AFBS; Cell Signaling Technologies, Danvers, MA) for 45 minutes at room temperature (RT). Avidin and Biotin were blocked using Vector Labs Avidin/Biotin blocking kit as per manufacturer's instructions (Vector Labs, Burlingame, CA). Slides were incubated overnight at 4˚C with the indicated antibody in AFBS. The following day, Vectastain ABC HRP Kit was used according to manufacturer's

instructions (Vector Labs). Biotinylated secondary antibody was prepared in AFBS and slides were incubated for 30 mins at RT. Slides were then developed with NovaRed substrate (Vector Labs) and lightly counterstained with Mayer's hematoxylin (Sigma Aldrich, St. Louis, MO). Slides were imaged using an Olympus BX43 microscope. The % of TG neurons stained by the respective antibodies were counted in a blinded fashion. Animal experiments were performed in duplicate at separate times for reproducibility.

## Examining the effects of Wnt/β-catenin antagonists on explant-induced reactivation

Mice infected for $\geq$ 30 days were operationally defined as being latently infected. TG from latently infected mice were minced into 3–4 pieces each and explanted into 6-well dishes in MEM that contained 10% FBS and incubated at 37˚C, 5% $CO_2$. Certain TG explants were incubated with 10 µM water soluble DEX (Sigma, D2915). Where indicated, 25 µM iCRT14 (Santa Cruz Biotechnology, CAS: 677331-12-3) or 10µM KYA1797K (Selleckchem, CAS: S8327) reconstituted in DMSO was added to TG explants. Each day 100µL media was removed and used to perform plaque assays on Vero cell monolayers as previously described [31].

## Statistical analysis

All graphs and comparisons were performed using GraphPad Prism software (v7.0d). *P* values less than 0.05 were considered significant for all calculations.

## Results

### More neurons express β-catenin in TG of mice latently infected with wt HSV-1

The Wnt signaling pathway enhances HSV-1 productive infection in cultured cells [32] and Wnt-associated genes are differentially expressed during the BoHV-1 latency-reactivation cycle [23]. Specifically, the transcription factor β-catenin is expressed in more TG neurons of latently infected calves when compared to TG neurons of uninfected calves or calves treated with DEX to induce viral reactivation [22–24]. In humans, the Wnt pathway has 19 genes encoding ligands and 15 receptor- or coreceptor-encoding genes [33]. It is not known which ligand activates each receptor; nor is it known which Wnt ligands or receptors are expressed in TG neurons. When a Wnt ligand engages its receptor, β-catenin protein levels generally increase, which was the rational for examining β-catenin as a marker for active canonical Wnt signaling. For these studies, we examined the effect of a LAT$^{-/-}$ mutant (dLAT2903) relative to wild-type (wt) HSV-1. LAT is the only viral gene product abundantly expressed in latently infected neurons and thus was an important comparison to wt LAT$^{+/+}$ HSV-1 [34]. The dLAT2903 mutant virus used for this study contains a deletion from -161 to +1667 relative to the start of the primary 8.3-kb LAT: consequently, dLAT2903 does not express detectable levels of LAT, and reactivation from latency in rabbits and mice is significantly reduced [34–37]. The LAT$^{-/-}$ mutant (dLAT2903) was marker repaired back to wt (dLAT2903R) and has identical properties as the parental wt McKrae strain of HSV-1 [34, 35]: therefore, dLAT2903R is referred to as wt HSV-1 throughout this study.

When compared to TG from uninfected mice (Fig 1; panel denoted as U), β-catenin+ TG neurons were readily detected in mice latently infected with wt HSV-1(WT panels; black arrows denote β-catenin+ TG neurons). Strikingly, β-catenin+ TG neurons were not readily detected in mice latently infected with dLAT2903 (panels denoted LAT$^{-/-}$). When IHC was performed with TG sections from mice latently infected with wt HSV-1, β-catenin+ TG

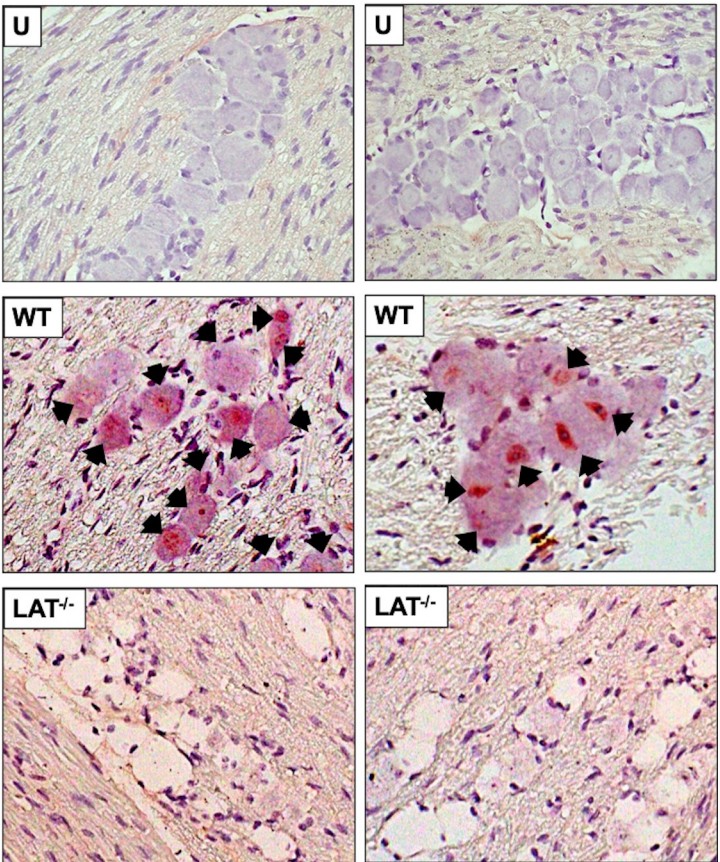

**Fig 1. β-catenin expression is increased in latently infected TG.** TG were harvested from uninfected mice (U panel) or mice latently infected with either wt HSV-1 or dLAT2903, a LAT-null mutant (LAT⁻/⁻). TG were dissected, formalin fixed, and paraffin embedded (n = 5/group). IHC was performed as described in the methods and materials using a rabbit polyclonal CTNNB1 (β-catenin) primary antibody (LifeSpan Bio, LS-C31415, diluted 1:250). Arrows denote β-catenin positive neurons. Sections were imaged at 600x magnification. Images are representative of two independent animal experiments.

neurons were not observed if the primary antibody was omitted (Fig 2A). Significantly more β-catenin+ TG neurons were detected in mice latently infected with wt HSV-1 when compared to TG sections prepared from mice latently infected with dLAT2903 and uninfected mice (Fig 2B). Approximately 60% of TG neurons were β-catenin+ compared to less than 20% in TG from mice latently infected with dLAT2903 and uninfected mice.

While LAT expression appears to be critical for increased β-catenin staining in TG neurons during latency, reduced establishment of dLAT2903 versus wt McKrae may also influence the % of TG neurons that were stained by the β-catenin antibody. For example, ocular infection of C57Bl/6 mice (no corneal scarification) with LAT⁺/⁺ McKrae establishes latency 3–4 times more efficiently relative to dLAT2903, as judged by viral DNA in TG [17]. Male Swiss Webster mice infected via ocular infection after corneal scarification with LAT⁺/⁺ 17syn+ HSV-1 virus strain (17-AHR1) establishes latency approximately 4 times more efficiently compared to a LAT deletion mutant that lacked 1,964 bp encompassing the LAT promoter and first 828 bp of the 5' end of LAT (17-AH) in [13]. Female Swiss Webster mice infected via the ocular cavity (no corneal scarification) with wt McKrae established latency approximately 2 times more efficiently than dLAT2903 (data not shown). In summary, the inability of dLAT2903 to establish

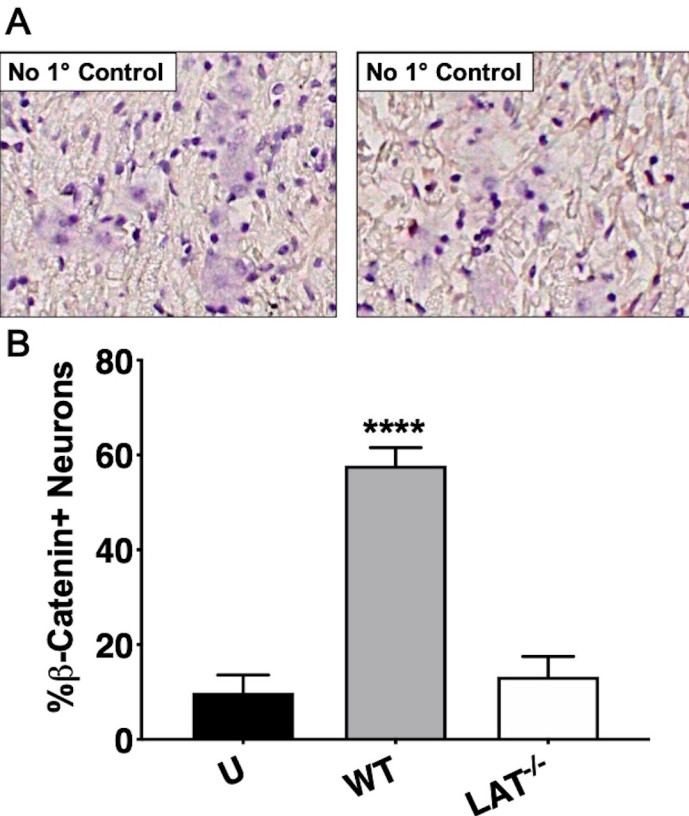

**Fig 2. Quantification of β-catenin expression.** A) IHC on latently infected TG was performed with the primary antibody omitted as a control. B) β-catenin positive TG neurons were counted from approximately 500 total neurons from two separate animal experiments. Data are shown as mean ± SEM. Asterisks denote significant differences calculated by students t-test, ****$p < 0.0001$.

latency as efficiently as wt McKrae (dLAT2903R) influenced the % of β-catenin+ TG neurons during latency: however, LAT expression clearly correlated with high levels of β-catenin expression in TG neurons during latency.

## Examination of β-catenin expression in TG during explant-induced reactivation

To test whether β-catenin expression was altered during viral reactivation from latency, TG from mice latently infected with HSV-1 (wt and dLAT2903) were explanted. TG explants were incubated with MEM + 2% charcoal-stripped FBS in the presence or absence of DEX. FBS passed through a column containing "activated" charcoal removes hormones, lipid-based molecules, certain growth factors, and cytokines yielding stripped FBS. However, this process does not remove salts, glucose, and most amino acids. FBS contains bio-active corticosteroids because nearly all Neuro-2A cells incubated with MEM + 10% FBS [38] contain glucocorticoid receptor (GR) in the nucleus. Conversely, nearly all Neuro-2A cells contain GR in the cytoplasm when incubated with 2% stripped FBS [38]. Furthermore, corticosterone levels were significantly higher in FBS compared to stripped FBS [31]. Recent studies demonstrated explant-induced reactivation from latency is stimulated by the synthetic corticosteroid DEX and expression of viral regulatory proteins were expressed earlier when TG explants were incubated with DEX [31]. At 4 or 8 hours after TG explant without DEX, β-catenin+ neurons were

readily detected in mice latently infected with wt HSV-1 (Fig 3; panels denoted WT 4 or WT 8). When TG explants were incubated with media containing DEX, β-catenin+ TG neurons were not frequently detected at 4 or 8 hours after explant (panels denoted WT 4+DEX or WT 8+DEX).

TG from mice latently infected with dLAT2903 were explanted and examined for β-catenin protein expression by IHC. In contrast to the results with wt HSV-1, DEX treatment increased the number of β-catenin+ TG neurons at 4 hours after explant from mice latently infected with dLAT2903 (Fig 4; panels LAT$^{-/-}$ 4 and LAT$^{-/-}$ 4+DEX). At 8 hours after TG explant, weak staining of TG neurons was detected by the β-catenin antibody in the absence of DEX treatment whereas DEX treatment appeared to increase the number of β-catenin+ TG neurons (panels denoted LAT$^{-/-}$ 8 and LAT$^{-/-}$ 8+DEX). These studies also revealed β-catenin+ staining was dispersed outside of TG neurons at 8 hours after TG explant when DEX was added to MEM. Published studies concluded that the Wnt pathway is activated within the central nervous system when injured, including surrounding non-neuronal cell types such as oligodendrocytes and microglia [39, 40], which suggests similar events may occur following explant of mice latently infected with dLAT2903.

As a control to TG from latently infected mice, β-catenin expression was performed in TG of uninfected mice. Following explant, β-catenin+ TG neurons were not readily detected at 4 hours after explant (panel U 4), but a few TG neurons were weakly stained 8 hours after explant (panel U 8; Fig 5). When DEX was added to cultures, more TG neurons from uninfected mice were weakly stained compared to TG explants incubated without DEX, or TG explants from mice latently infected with wt HSV-1 (Fig 2). Furthermore, the intensity of β-catenin staining in uninfected TG was not as strong relative to TG of mice latently infected with wt HSV-1 (Fig 3) suggesting more β-catenin was expressed in TG neurons of mice latency infected with wt HSV-1. Finally, we noted that following TG dissection, explant, and incubation in medium TG neurons in sections were not as pristine compared to when they were dissected and immediately fixed in formalin.

The number of β-catenin+ TG neurons in the respective samples were quantified and these results are shown in Fig 6. TG from mice latently infected with wt HSV-1 contained similar levels of β-catenin+ TG neurons at 4 and 8 hours after TG explant (approximately 60% were β-catenin+; Fig 6A). When DEX was added to TG explants from mice latently infected with wt HSV-1, significantly fewer TG neurons were β-catenin+. At 4 hours post-explant (no DEX treatment), TG from mice latently infected with dLAT2903 contained significantly more β-catenin+ TG neurons relative to latency (Fig 6B). Interestingly, mice latently infected with dLAT2903 had approximately 60% β-catenin+ TG neurons when explants were treated with DEX for 4 hours, which was similar to TG from mice latently infected with wt HSV-1. At 8 hours post-explant, β-catenin expression decreased to levels similar to mice latently infected with dLAT2903 (~10%, Fig 6B); however, DEX treatment increased β-catenin expression 3-fold. Overall, TG from uninfected mice was more similar to TG from mice latently infected with dLAT2903 than wt HSV-1 (Fig 6C). While the intensity of β-catenin expression in uninfected mice (Fig 5) was clearly less than latently infected TG (Figs 3 and 4), approximately 80% of TG neurons were β-catenin+ at 4 hours after explant when incubated in MEM containing DEX (Fig 6C; 4 + DEX). At 8 hours after explant in the presence of DEX, the number of β-catenin+ TG neurons was reduced to approximately 60%. In summary, these studies indicated explant-induced reactivation in the presence of DEX dramatically altered β-catenin expression in TG neurons. While these studies suggested LAT expression influenced β-catenin expression during explant-induced reactivation, published reports concluded RNA encoding ICP0, ICP4, and/or thymidine kinase can be detected in TG of mice latently infected with HSV-1 [41, 42].

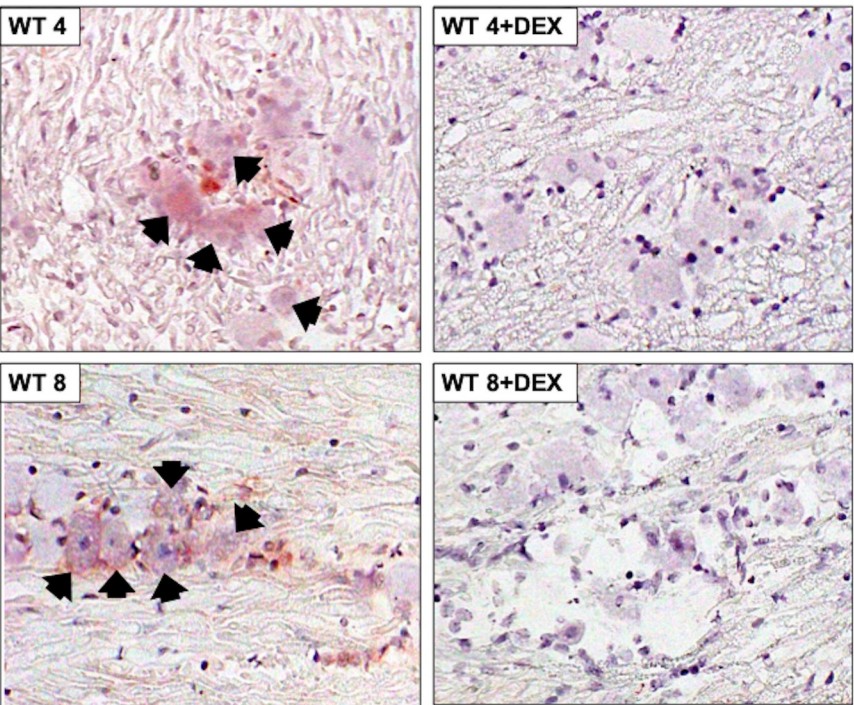

**Fig 3. β-catenin protein expression in TG explants of WT latently infected mice.** TG from wt HSV-1 latently infected mice were explanted 4 or 8 hours with or without DEX, subsequently formalin fixed and paraffin embedded. TG thin sections were stained with a β-catenin antibody as described in Fig 1. Arrows denote β-catenin positive neurons and magnification of sections is 600x. Images are representative of two independent animal experiments.

If these lytic cycle proteins are expressed, they may have differential impacts on β-catenin expression, depending on whether LAT is expressed.

## Antagonists of the Wnt-signaling pathway significantly reduced DEX-induced virus reactivation from latency in TG explants

The data above revealed β-catenin was differentially expressed during the latency-reactivation cycle. However, these studies did not address whether the Wnt/β-catenin signaling pathway influenced virus shedding during explant-induced reactivation. To examine the effect the Wnt/β-catenin signaling pathway had on explant-induced reactivation from latency, two different Wnt-pathway antagonists were used: iCRT14 and KYA1797K. iCRT14 is a thiazolidine-dione inhibitor that impairs β-catenin binding with a T-cell factor (TCF) family member and subsequent TCF binding to DNA: consequently, β-catenin dependent transcription is not induced [43]. KYA1797K enhances formation of the β-catenin destruction complex through binding of Axin, which stimulates GSK3β activation [44]. For each antagonist, cytotoxicity studies using both Vero cells and a mouse neuroblastoma cell line (Neuro-2a, ATCC CCL131) were performed, showing no effect with concentrations up to 50μM [32] and data not shown.

   TG from mice latently infected with wt HSV-1 were explanted in media containing 10% FBS and either iCRT14, KYA1797K or DMSO (Fig 7). Each day after TG explant, 100μL of the supernatant was removed and used to measure levels of reactivated virus. Relative to the vehicle control, a 2-3-log decrease was detected in cultures treated with either antagonist (Fig 7), and this trend continued through 9 days post-explant. At early timepoints, KYA1797K was not as effective compared to iCRT14 (approximately $10^3$ vs $10^1$, respectively). However, both

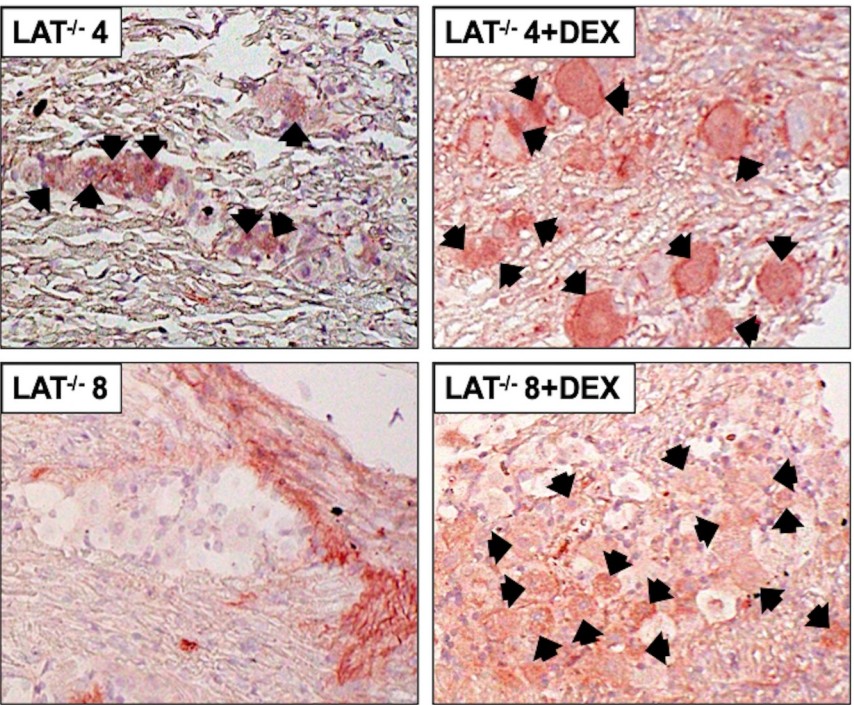

**Fig 4. β-catenin protein expression in TG explants of dLAT2903 latently infected mice.** TG from mice latently infected with a LAT-null HSV-1 (LAT$^{-/-}$) were either collected at ≥30 days after infection (latency) or explanted 4 or 8 hours in MEM containing 2% stripped FBS with or without DEX, subsequently formalin fixed and paraffin embedded. TG thin sections were stained with a β-catenin antibody as described in Fig 1. Arrows denote β-catenin positive neurons. Magnification of sections is 600x. Images are representative of two independent animal experiments.

exhibited a plateau in virus shedding around day 7 at ~10$^3$ PFU from TG explants. TG explants incubated with just FBS exhibited peaked levels of virus shedding at day 6 with 10$^6$ PFU/mL and remained at this level until day 9, which was consistent with previous studies [31].

Given the most dramatic effect was observed using iCRT14 at 5 days post-explant, this antagonist and timepoint were used to analyze the effect on reactivation of dLAT2903 infected TG. Since dLAT2903 displays significantly reduced reactivation following explant, latently infected TG were explanted in both 10% FBS or 2% stripped serum + DEX to compare virus reactivation [31]. Regardless of the treatment, virus reactivation was significantly reduced when compared to WT, ranging between 10$^2$ to 10$^{3-}$ PFU/mL (Fig 8). Upon addition of 25μM iCRT14, virus shedding was reduced approximately 2-log when TG were explanted in 10% FBS (<10$^2$); but, was not detected when explanted in 2% stripped serum + DEX (Fig 8, N.D: none detected). Collectively, this data demonstrated that the Wnt/β-catenin signaling pathway enhanced explant-induced reactivation from latency in mice latently infected with wt HSV-1 and the LAT$^{-/-}$ mutant.

## Discussion

These studies revealed that more β-catenin+ TG neurons and higher levels of β-catenin were detected in mice latently infected with wt HSV-1 relative to TG from uninfected mice or mice latently infected with a LAT$^{-/-}$ mutant suggesting LAT, in part, plays a role in mediating β-catenin expression during latency. It is assumed LAT does not encode a functional protein that regulates the latency-reactivation cycle [3]. Hence, the dogmatic belief is small non-coding RNAs mediate key events during the latency-reactivation cycle, including micro-RNAs located

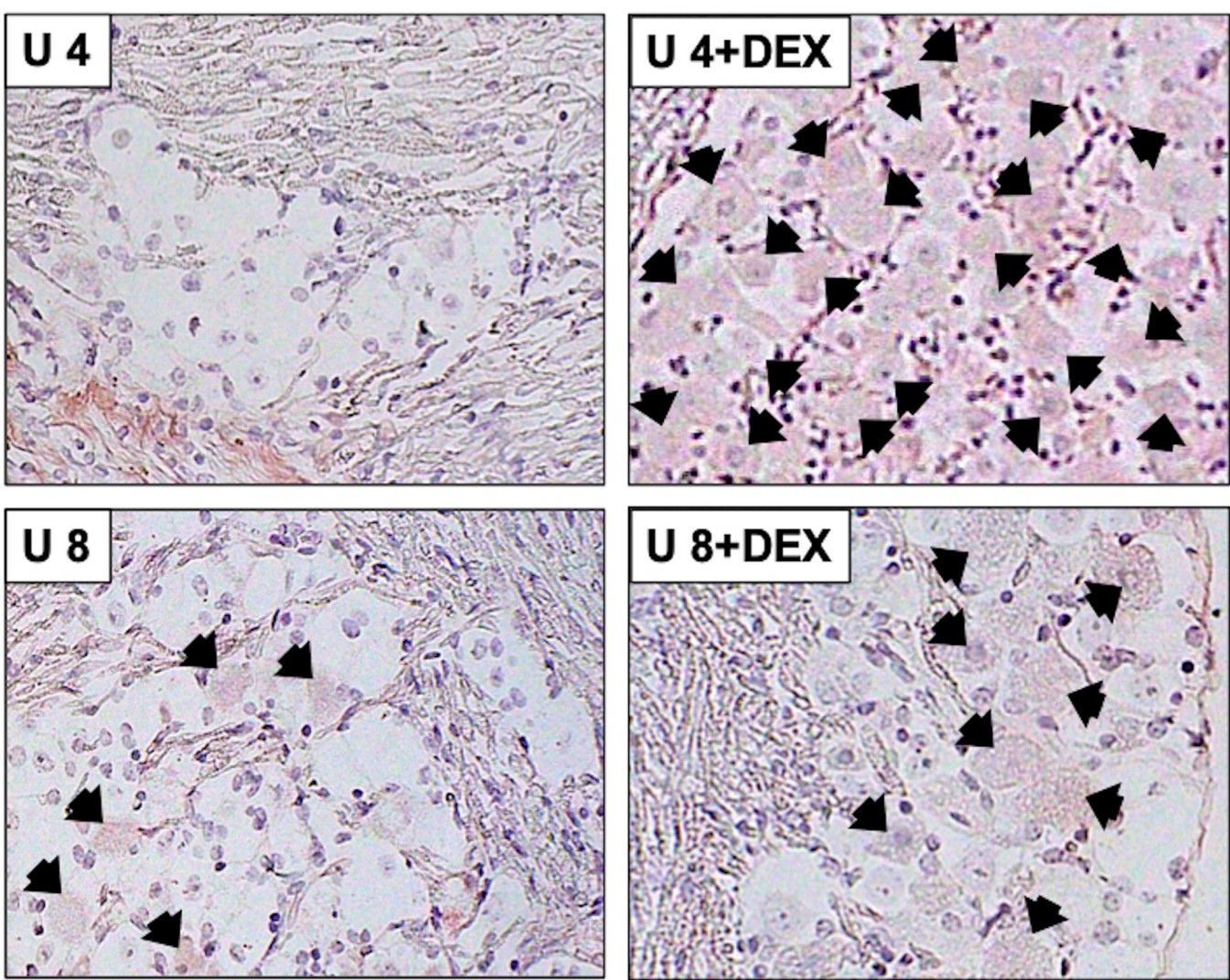

**Fig 5. β-catenin protein expression in TG explants of uninfected mice.** TG from uninfected mice were explanted 4 or 8 hours with or without DEX, subsequently formalin fixed and paraffin embedded. TG thin sections were stained with a β-catenin antibody as described in Fig 1. Arrows denote β-catenin positive neurons and magnification of sections is 600x. Images are representative of two independent animal experiments.

within the LAT locus [6, 7]. Certain LAT encoded miRNAs interfere with expression of viral transcripts that encode key regulatory proteins. For example, miR-H3 [6] interferes with expression of the key viral transcriptional regulator ICP4. ICP34.5, an important lytic neuro-virulence factor, is targeted by two viral miRNAs, miR-H3 and miR-H4 [6]. A mutant virus that does not express the LAT-encoded miR-H2 has increased neurovirulence and reactivation [45], in part because this miRNA interferes with ICP0 expression [6, 46]. Two small non-coding RNAs, which are not micro-RNAs, are expressed in TG of mice latently infected with wt HSV-1; interestingly, sequences encoding these small non-coding RNAs are deleted from dLAT2903 [8]. These LAT small non-coding RNA interfere with apoptosis and ICP4 expression [47] suggesting they also play a role in the latency-reactivation cycle. Analysis of the known miRNAs encoded by the human herpesviruses are reported to preferentially target the Wnt signaling pathway [48], which supports the premise that stabilizing β-catenin expression in TG neurons during latency is important for maintaining a life-long latent infection in sensory neurons.

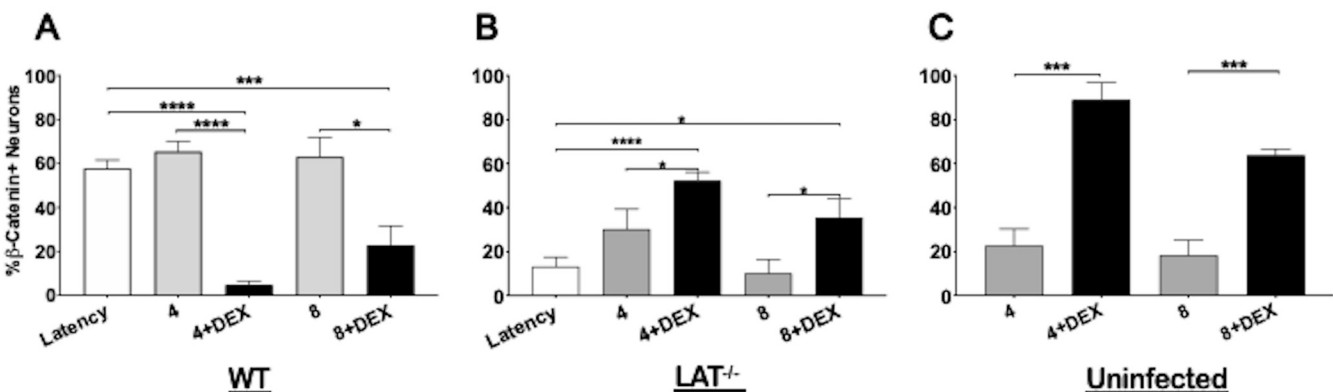

**Fig 6. Quantification of β-catenin expression in explanted TG.** TG from WT (A), LAT-null mutant (LAT[-/-], B) or uninfected (C) mice were explanted in MEM containing 2% stripped serum for the indicated times with or without DEX. β-catenin+ neurons were calculated from approximately 500 total neurons from two independent animal experiments. Data are shown as mean ± SEM. Asterisks denote significant difference calculated by students t-test, $^*p<0.05$, $^{***}p<0.001$, $^{****}p<0.0001$.

We estimated that approximately 60% of TG neurons expressed β-catenin in mice latently infected with wt McKrae. A previous report concluded that approximately 30% of TG neurons in mice were latently infected with a virulent HSV-1 strain [49] suggesting a subset of β-catenin+ TG neurons were not infected. LAT miRNAs (miR-H3, miR-H5, and mIR-H6) are present in exosomes that are transported from productively infected cells to uninfected cells [50].

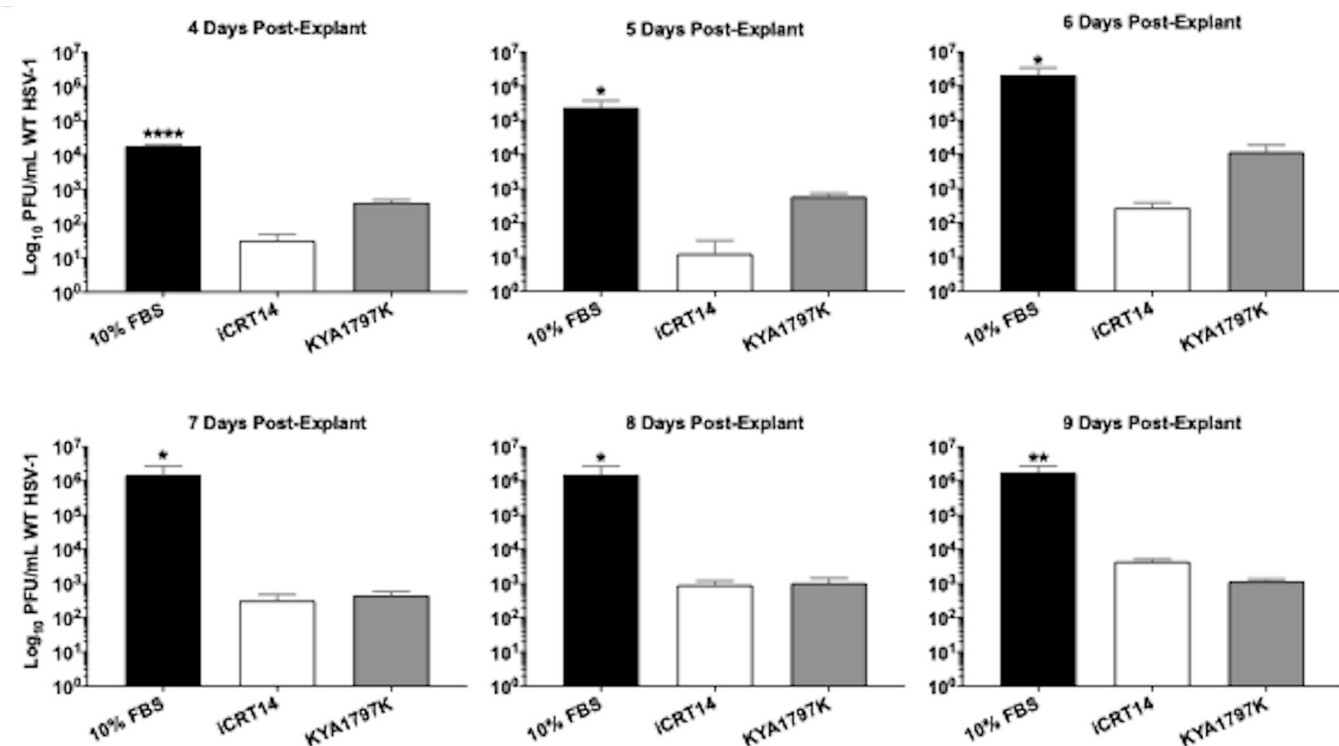

**Fig 7. Explant-induced reactivation from latency in the presence of Wnt antagonists.** Wt infected TG were harvested ~30dpi (latency) and explanted in 10% FBS with either 25μM iCRT14 or 10μM KYA1797K. Untreated controls (10% FBS) were included as reactivation controls. Daily, 100μL supernatant was removed and used to plaque for viral PFU/mL. Data are shown as mean ± SD for quadruplicate wells. Asterisks denote significant difference calculated by students t-test, $^*p<0.05$, $^{**}p<0.005$, $^{****}p<0.0001$.

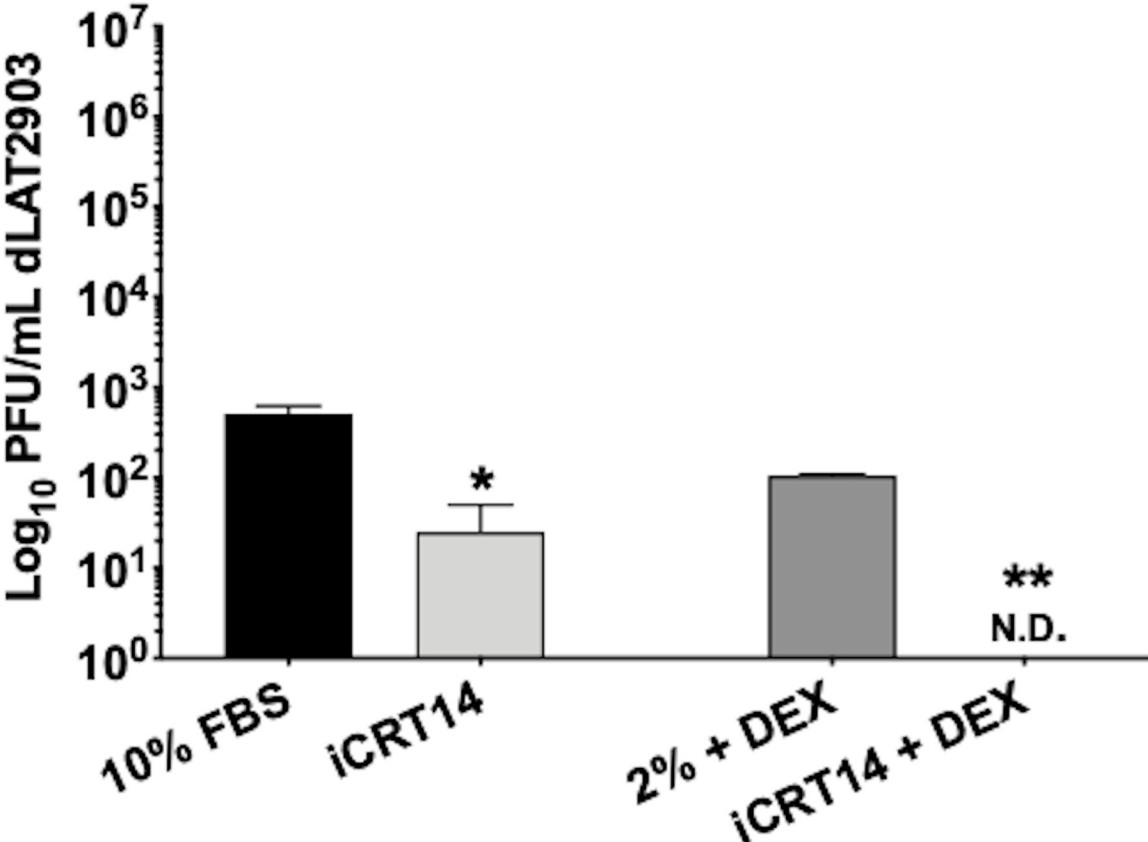

**Fig 8. TAiCRT14 inhibits explant-induced reactivation from latency in dLAT2903 infected TG.** Peak inhibition of WT virus reactivation was observed using 25μM iCRT14, 5 days post-explant; therefore, this antagonist and time point were used to assess inhibition of LAT$^{-/-}$ (dLAT2903) reactivation. LAT$^{-/-}$ infected TG were harvested 30dpi (latency) in either 10% FBS or 2% stripped FBS with DEX and 25μM iCRT14. Untreated controls (either 10% FBS only or 2% stripped serum with DEX) were included as positive controls for reactivation. 5 days post-explant, 100μL supernatant was removed and used to measure viral shedding (PFU/mL). Data are shown as mean ± SD for duplicate wells. Asterisks denote significant difference calculated by students t-test, $^*p<0.05$, $^{**}p<0.005$; N.D. = None detected.

Hence, it is tempting to suggest exosomes containing LAT gene products are transported to surrounding uninfected TG neurons or non-neuronal support cells: consequently, β-catenin expression is activated in "bystander" cells, including neurons. However, it should be pointed out there is no current published study showing exosomes frequently transport LAT non-coding RNAs to uninfected TG neurons during latency. Extracellular factors regulate (positively as well as negatively) the Wnt/β-catenin signaling pathway [33, 51] suggesting mice latently infected with wt HSV-1 produce a soluble factor that stimulates β-catenin expression or increases its ½ life in bystander cells. Strikingly, Wnt16, a soluble Wnt agonist, is expressed at 54 times higher levels in TG of calves latently infected with BoHV-1 when compared to TG from calves treated with DEX for 30 minutes to initiate reactivation from latency [23]. Regardless of how the Wnt/β-catenin signaling pathway is activated in TG of mice latently infected with wt HSV-1, activation of this signaling pathway would promote neuronal survival and neurogenesis [26, 39, 52, 53], including TG neurons that do not contain viral genomes.

In general, increased corticosteroids, due to stress and depression, interfere with β-catenin in the nervous system [54]. Furthermore, stress induces expression of soluble Wnt antagonists that interfere with β-catenin expression, and induces neuronal death [51, 55]. While the findings with wt HSV-1 are in concordance with these findings, it was clear β-catenin+ TG neurons from uninfected mice or mice latently infected with dLAT2903 increased following

explant. In particular, the number of β-catenin+ TG neurons in mice latently infected with dLAT2903 or uninfected mice increased when DEX was added. During explant-induced reactivation, the normal tropic support that TG receive in vivo is disrupted, which may impact how certain signaling pathways are regulated following explant. Additional evidence that explant-induced reactivation does not recapitulate all events during reactivation in vivo comes from a recent study that demonstrates apoptosis is induced in many TG neurons from latently infected mice within 4 hours after explant [31]. In contrast, TG neurons do not frequently undergo apoptosis during early stages of DEX induced reactivation in calves latently infected with BoHV-1 in vivo [56, 57]. It is also unlikely apoptosis occurs frequently in human TG neurons undergoing HSV-1 reactivation. While it seems clear that explant-induced reactivation in the mouse model has limitations, these studies indicated that LAT encoded products, directly or indirectly, influenced β-catenin expression in TG neurons during latency.

Since β-catenin was preferentially expressed during latency, it was surprising that inhibiting the Wnt/β-catenin signaling pathway consistently reduced virus shedding during explant-induced reactivation. However, iCRT14 reduced HSV-1 virus yields in productively infected non-neuronal cells (human fetal lung and Vero cells) and a late protein, VP16, enhanced β-catenin dependent transcription as well as stabilized β-catenin steady state protein levels [32]. An independent study revealed infected cells can be divided into transcriptionally unique subsets: furthermore, cells abortively infected were identified where antiviral signaling blunted productive infection [58]. In certain "highly infected cells" transcriptional reprogramming occurred, in part because β-catenin is recruited to the nucleus and viral replication compartments: hence, β-catenin promotes late viral gene expression and progeny production [58]. While the potential dual role for β-catenin during the latency-reactivation cycle may appear to be implausible, it is well established that the Wnt/β-catenin signaling pathway has different effects in a cell-type- and contextual-specific manner. For example, dysregulation of the Wnt/β-catenin pathway is crucial for development of certain types of cancer [59]: conversely, the same pathway mediates neurogenesis and cell survival in the nervous system [28, 29]. Based on these observations, we suggest that during establishment and maintenance of latency following infection with wt virus, β-catenin expression in TG neurons promotes neuronal survival [53] and neurogenesis [28, 29]. During early phases of reactivation in mice latently infected with wt virus, β-catenin may be recruited to replication complexes in certain latently infected TG neurons. This could be one of many transcriptional reprogramming events that culminates in stimulating lytic cycle viral gene expression and virus shedding, hallmarks of successful reactivation from latency.

While these studies suggested β-catenin plays a role during the HSV-1 latency-reactivation cycle, additional studies are necessary to unravel the mechanism by which 1) LAT regulates β-catenin expression during latency and 2) how β-catenin regulates the maintenance of latency but has a potential role during early stages of reactivation from latency.

## Author Contributions

**Conceptualization:** Kelly S. Harrison, Liqian Zhu, Clinton Jones.

**Data curation:** Kelly S. Harrison, Liqian Zhu, Clinton Jones.

**Formal analysis:** Kelly S. Harrison, Clinton Jones.

**Funding acquisition:** Kelly S. Harrison, Clinton Jones.

**Investigation:** Liqian Zhu, Prasanth Thunuguntla, Clinton Jones.

**Methodology:** Kelly S. Harrison, Clinton Jones.

**Project administration:** Clinton Jones.

**Resources:** Clinton Jones.

**Supervision:** Kelly S. Harrison, Clinton Jones.

**Validation:** Clinton Jones.

**Visualization:** Clinton Jones.

**Writing – original draft:** Clinton Jones.

**Writing – review & editing:** Kelly S. Harrison, Clinton Jones.

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
