## [Decision Letter · Decision Letter 0]

14 Jan 2020

PONE-D-19-33585

Herpes simplex virus 1 regulates beta-catenin expression in TG neurons during the latency-reactivation cycle.

PLOS ONE

Dear Dr Jones,

Thank you for submitting your manuscript to PLOS ONE. After careful consideration, we feel that it has merit but does not fully meet PLOS ONE’s publication criteria as it currently stands. Therefore, we invite you to submit a revised version of the manuscript that addresses the points raised during the review process.

The reviews are mixed, with reviewer #1 having no major issues and reviewer #2 having many.  Please address all the issues in the reviews, particularly the major issues of reviewer #2.

We look forward to receiving your revised manuscript. To enhance the reproducibility of your results, we recommend that if applicable you deposit your laboratory protocols in protocols.io, where a protocol can be assigned its own identifier (DOI) such that it can be cited independently in the future. For instructions see: http://journals.plos.org/plosone/s/submission-guidelines#loc-laboratory-protocols

We look forward to receiving your revised manuscript.

Kind regards,

Neal A. DeLuca, Ph.D.

Academic Editor

PLOS ONE

Journal Requirements:

2. In your Methods section, please give the sources of any viral strains and cell lines used in your study.

3. Thank you for including your ethics statement:

'Animal studies were reviewed by OSU IACUC committee and the studies were approved (VM15-26).'

a. Please amend your current ethics statement to include the full name of the ethics committee that approved your specific study.

For additional information about PLOS ONE submissions requirements for ethics oversight of animal work, please refer to http://journals.plos.org/plosone/s/submission-guidelines#loc-animal-research  

5. Please include captions for your Supporting Information files at the end of your manuscript, and update any in-text citations to match accordingly. Please see our Supporting Information guidelines for more information: http://journals.plos.org/plosone/s/supporting-information

Reviewers' comments:

Reviewer's Responses to Questions

**Comments to the Author**

1. Is the manuscript technically sound, and do the data support the conclusions?

Reviewer #1: Yes

Reviewer #2: No

2. Has the statistical analysis been performed appropriately and rigorously? 

Reviewer #1: Yes

Reviewer #2: Yes

3. Have the authors made all data underlying the findings in their manuscript fully available?

Reviewer #1: Yes

Reviewer #2: Yes

4. Is the manuscript presented in an intelligible fashion and written in standard English?

Reviewer #1: Yes

Reviewer #2: Yes

5. Review Comments to the Author

Reviewer #1: Review of: Herpes simplex virus 1 regulates beta-catenin expression in TG neurons during the latency-reactivation cycle

By Harrison, Zhu, Thunuguntla and Jones,

In this manuscript, the authors show that β-catenin is upregulated in TG neurons latently infected with HSV-1 in a LAT dependent manner. The authors further establish that induction of reactivation by dexamethasone treatment results in dramatic downregulation/degradation of β-catenin. This decrease was not observed in TG infected with LAT null virus. Furthermore, expression of β-catenin is beneficial for virus because inhibition of this pathway during reactivation results in reduced viral shedding.

The findings reported here are very interesting and valuable for the advancement of the field, and only minor improvements are suggested as described below.

1. Line 68: How soon after reactivation does Wnt signaling decrease? In figure 6, no decrease in β-catenin expression is seen at 4hr, 8hr post ex-plantation. Why were these timepoints selected?

2. Figures 1 and 2: Was β-catenin detected exclusively in neurons? Did you observe any immune cells/supporting cells expressing β-catenin?

3. Line 200: Word “estimated” is a bit unclear- did you count number of β-catenin positive neurons over total number of neurons to determine percentage or “eyeball” percentage of β-catenin positive neurons?

4. Figure 4: LAT-/- 4 panel: cells surrounding β-catenin positive cells look different from other panels. Also, not necessary but if you can please replace it with a better image.

5. Line 212- What does “GR” stand for?

6. Line 235 states that β-catenin staining was detected outside of TG neurons. Can you show this in Fig 4 and elaborate on the significance of this finding?

7. Line 251: Authors state that intensity of β-catenin was lower in uninfected TG than in infected TG. This suggests that while higher percentage of cells might stain positive for β-catenin, the overall expression levels may be lower. Could you elaborate on this?

8. Figure 6: Dex treatment in uninfected TG results in roughly two-fold higher β-catenin positive neurons when compared to LAT-/-, dex treated TG. These results suggest that viral factors other than LAT may contribute to β-catenin downregulation. Can you please comment on this possibility?

9. The finding that the percentage of cells expressing β-catenin is higher than the reported number of infected cells is intriguing, as it suggests that other, surrounding uninfected cells are also producing β-catenin. The authors propose this could be caused by miRNA/sncRNA encoded by LAT that are released in exosomes. Perhaps the authors could speculate the functional significance of causing β-catenin upregulation in surrounding cells (is it beneficial to the virus to have surrounding cells overexpress β-catenin?).

10. It seems the two sncRNAs may play a more important role in latency-reactivation than miRNA based on the authors work and Wechsler’s work. Perhaps the authors could elaborate on this?

11. Figure 7 suggests that inhibition of Wnt signaling results in decreased viral replication. This seems counterintuitive considering that Wnt pathway is turned off/ β-catenin expression is decreased during reactivation (as stated on lines 68)? Perhaps the authors could elaborate on this?

12. The authors are a bit cautious in their discussion, and could expand the discussion section a bit if they wish.

Reviewer #2: Harrison and colleagues have investigated the expression of beta-catenin in the context of herpes simplex virus 1 latency and reactivation using a mouse ocular infection model. The authors determined that beta-catenin expression was enhanced in trigeminal ganglia (TG) of mice latently infected with a wild type (WT) HSV-1 strain compared to uninfected animals, or animals latently infected with an HSV-1 LAT deletion mutant. Reactivation from latency was achieved by explanting latently infected TG in the presence of dexamethasone (DEX). Whereas beta-catenin expression was reduced in TG latently infected with WT virus upon explant in the presence of DEX, beta-catenin expression was significantly enhanced in explanted TG from both LAT infected and uninfected animals that had been treated with DEX. In a final set of experiments the authors show that compounds that inhibit the Wnt/beta-catenin signalling pathway are potent inhibitors of WT HSV-1 reactivation in explanted TG. The authors central conclusions are that the Wnt/beta-catenin signalling pathway is important for HSV-1 reactivation and that beta-catenin is differentially expressed during latency (high expression) and reactivation (low expression) and that LAT somehow regulates beta-catenin expression.

While the findings are potentially interesting, the study is incomplete, inadequately controlled, and the data are of variable quality. Thus, the findings are too preliminary to be particularly informative to researchers in this field.

Major issues:

1) Can the authors confirm that TG neurons are latently infected and quantify this? Are the TG neurons expressing beta-catenin the ones that are latently infected? What proportion of latently infected TG neurons express beta-catenin?

2) The uninfected controls in figure 5 do not look like the uninfected control sections in figure 1. Why? The uninfected TG explanted in DEX in figure 5 that are highlighted with arrowheads do not look to be beta-catenin positive. If the primary antibody was excluded in control experiments and the samples double-blinded could beta-catenin positive cells be identified with confidence?

3) Figure 6 and sentence on lines 347-349. Authors should include uninfected TG from mice that had not been explanted (i.e. fixed immediately after removal) to control for the effects of explant on beta-catenin expression.

4) Sentence on lines 279-280. What are the numbers of latently infected TG neurons in animals infected with WT virus compared to LAT mutant virus? Need to show this important control.

5) Figure 6. What is the degree of reactivation from TG explanted from WT and LAT mutant infected animals? Presumably, the medium associated with the explanted TG analyzed for beta-catenin expression shown in figure 6 was available for virus titration. 

6) Figure 7. How do Wnt/beta-catenin inhibitors impact the reactivation of LAT mutants? How would DEX impact the action of the Wnt/beta-catenin inhibitors on virus reactivation?

7) Lines 341-343. This is an egregious over-interpretation of the data.

Minor issues:

1) It is curious that Wnt/beta-catenin agonists are so potent at inhibiting reactivation when the levels of beta-catenin are so low during the reactivation phase. Can the authors comment on this apparent inconsistency?

2) Line 27. Why highlight ocular disease here? Line 22 introduces ocular, nasal and oral infection. Consider "recurrent disease".

3) Line 35. "increased" compared to what?

4) Line 47. Odd phrase. Suggests oral disease upon reactivation after primary infection of the eye? This is confusing. Similar issues in abstract. Consider rephrasing.

5) Lines 49-50. This sentence implies LAT is not expressed during lytic infection. I don't think this is the point the authors are trying to make. Consider rephrasing.

6) Line 53. Why does the nature of LAT locus products "suggest" functions related to latency? Consider rephrasing.

7) Line 76. “Wnt binding” to what?

8) Line 170. “rational” should be “rationale”.

9) Line 177. “marker rescued” should read “repaired”.

10) Sentence on lines 308-309. Alternatively, the data may indicate that the inhibitors were toxic to the explants. Can the authors control for this? Is there a dose response to the inhibitors?

6. PLOS authors have the option to publish the peer review history of their article (what does this mean?). If published, this will include your full peer review and any attached files.

Reviewer #1: No

Reviewer #2: No

---

## [Author Response · Author response to Decision Letter 0]

2 Mar 2020

With respect to the comments of Reviewer #1, the following changes in the revised manuscript were made.

Concern #1: Line 68: How soon after reactivation does Wnt signaling decrease? In figure 6, no decrease in β-catenin expression is seen at 4hr, 8hr post explantation. Why were these timepoints selected? 

Response: �-catenin+ TG neurons were nearly undetectable after TG explants were incubated with MEM, 2% stripped FBS, and DEX. However, �-catenin+ TG neurons were readily detectable when there was no DEX added to TG explants. When TG explants are treated with DEX to stimulate reactivation from latency, there are significantly more caspase 3+ TG neurons at 4, 8 and 16 hours after TG explant relative to TG explants not treated with DEX, Refence #31. Since there was a clear-cut difference in �-catenin expression during early times after TG explant in the presence of DEX, we did not examine �-catenin expression at later times because of increased apoptosis. 

Concern #2: Figures 1 and 2: Was β-catenin detected exclusively in neurons? Did you observe any immune cells/supporting cells expressing β-catenin? 

Response: For mice latently infected with wt HSV-1, most β-catenin staining was detected in neurons. In TG explants from mice latently infected with dLAT2903, we saw staining observed in neurons and other cells, which is pointed out in lines 253-258 (Figure 4). Since we did not include any differential stain to establish what types of immune/supporting cells were expressing, it is not clear which cells types are expressing β-catenin.

Concern #3: Agreed. Line 200: Word “estimated” is a bit unclear- did you count number of β-catenin positive neurons over total number of neurons to determine percentage or “eyeball” percentage of β-catenin positive neurons? 

Response: We have replaced the word “estimated” with “counted”; total number of β-catenin positive neurons over total number of neurons to determine percentage because we did in fact count �-catenin+ TG neurons (lines 147-148 and 217-218). 

Concern #4: Figure 4: LAT-/- 4 panel: cells surrounding β-catenin positive cells look different from other panels. Also, not necessary but if you can please replace it with a better image. 

Response: A higher quality image has been used for LAT-/- panel in Figure 4.

Concern #5: Line 212- What does “GR” stand for? 

Response: GR is glucocorticoid receptor, and this was defined in the revised manuscript (line 228-230). We have carefully examined the manuscript to ensure all acronyms have been defined.

Concern #6: Line 235 states that β-catenin staining was detected outside of TG neurons. 

Response: Recent published studies demonstrated the Wnt pathway is activated within the CNS after injury, including surrounding non-neuronal cell types such as oligodendrocytes and microglia. Thus, the observation of β-catenin staining outside of neurons may be due to expression in supportive cells. This point is included in the revised manuscript (lines 255-258).

Concern #7: Line 251: Authors state that intensity of β-catenin was lower in uninfected TG than in infected TG. This suggests that while higher percentage of cells might stain positive for β-catenin, the overall expression levels may be lower. Could you elaborate on this? 

Response: This statement was made because the intensity of staining did not appear to be as bright, which is difficult to quantify. However, it was clear staining was brighter, which suggested there is more β-catenin in these samples. These points were clarified in the revised manuscript (lines 273-276). 

Concern #8: Figure 6: Dex treatment in uninfected TG results in roughly two-fold higher β-catenin positive neurons when compared to LAT-/-, dex treated TG. These results suggest that viral factors other than LAT may contribute to β-catenin downregulation. Can you please comment on this possibility?

Response: There have been reports that additional viral gene products can be detected in TG during latency; ICP0, ICP4, and thymidine kinase RNA for example. If these lytic cycle regulatory proteins are expressed (ICP0 and ICP4 in particular) they could influence cellular gene expression. This point is included in the revised manuscript (lines 303-309). 

Concern #9: The finding that the percentage of cells expressing β-catenin is higher than the reported number of infected cells is intriguing, as it suggests that other, surrounding uninfected cells are also produce β-catenin. The authors propose this could be caused by miRNA/sncRNA encoded by LAT that are released in exosomes. Perhaps the authors could speculate the functional significance of causing β-catenin upregulation in surrounding cells (is it beneficial to the virus to have surrounding cells overexpress β-catenin?)

Response: We have expanded on the possibility of LAT-encoded non-coding RNAs being transported to surrounding TG neurons via exosomes and why this may impact surrounding neurons, and perhaps non-neuronal cells during latency (lines 393-410). 

Concern #10: It seems the two sncRNAs may play a more important role in latency-reactivation than miRNA based on the authors work and Wechsler’s work. Perhaps the authors could elaborate on this?

Response: We have expanded this part of the discussion, as suggested (lines 372-391). 

Concern #11: Figure 7 suggests that inhibition of Wnt signaling results in decreased viral replication. This seems counterintuitive considering that Wnt pathway is turned off/ β-catenin expression is decreased during reactivation (as stated on lines 68)? Perhaps the authors could elaborate on this? 

Response: This section has been revised to explain how �-catenin may have dual roles during the latency-reactivation cycle. (lines 430-451).

Concern #12: The authors are a bit cautious in their discussion and could expand the discussion section a bit if they wish.

Response: After addressing Concerns 9-11, I think we have speculated as much as I feel comfortable until we pursue this project more in detail. 

With respect to the comments of Reviewer #2, the following changes were made.

Concern #1: Can the authors confirm that TG neurons are latently infected and quantify this? Are the TG neurons expressing beta-catenin the ones that are latently infected? What proportion of latently infected TG neurons express beta-catenin? 

 Response: In the discussion, we proposed two mechanisms by which �-catenin may be expressed in latently infected TG neurons as well as a subset of TG neurons that are “bystanders” (lines 393-410). It is not trivial to accurately calculate the % of TG neurons that are latently infected and express ��catenin. To do this, one would have to cut consecutive sections and perform IHC on one section to identify �-catenin+ TG neurons and on the other section perform IHC to identify a viral protein during reactivation. We tried this approach and it was difficult to interpret because during explant-induced reactivation �-catenin+ TG neurons are nearly undetectable. It is possible to identify LAT+ TG neurons using in situ hybridization on one section; on the consecutive section perform IHC to visualize �-catenin+ TG neurons. There are a couple of issues with this approach, which would make it difficult to achieve a clear-cut and accurate result. First, it is well established there are HSV-1 genome+ TG neurons that do not express detectable levels of LAT during latency in mice: however, it is not clear what % of the total latently infected TG neurons do not express LAT. Secondly, TG sections from latently infected mice are very difficult to obtain superior consecutive sections. We routinely obtain excellent consecutive section from calves latently infected with bovine herpesvirus 1, but not mice latently infected with HSV-1. Finally, it is difficult to use two antibodies that are differentially stained on the same TG section, at least in our hands. We have done this with brain tissue, but when the same antibodies were used for TG sections the background was terrible. In summary, I do not believe these studies would yield convincing or accurate results. If necessary, these points can be added to the revised manuscript.

Concern #2: The uninfected controls in figure 5 do not look like the uninfected control sections in figure 1. Why? 

Response: The uninfected sections presented in Figure 1 were dissected as intact TG and then directly placed into formalin post-euthanasia. Uninfected tissue in Figure 5 was minced, explanted in MEM containing 2% stripped FBS, and then incubated for 4 and 8 hours. These small TG pieces were then collected and placed in formalin and paraffin embedded. While the integrity of the tissue is not as good as intact TG immediately placed in formalin, they clearly show the general morphology of TG. This is stated in the revised manuscript (lines 276-278).

Concern #3: The uninfected TG explanted in DEX in figure 5 that are highlighted with arrowheads do not look to be beta-catenin positive. If the primary antibody was excluded in control experiments and the samples double-blinded could beta-catenin positive cells be identified with confidence? 

Response: The TG neurons in Figure 5 are stained (albeit weakly) relative to the no primary antibody control. The no primary antibody control was performed using TG sections form mice latently infected with wt HSV-1: samples from explanted TG (no primary antibody control) also do not yield any staining: this is stated in the revised manuscript (lines 271-276). if necessary, we can add the no primary antibody TG sections for uninfected TG that was explanted to Figure 5. All TG sections in this study were analyzed in a blind fashion. This was stated in the materials and methods of the original manuscript (lines 147-148). 

Concern #4: Figure 6 and sentence on lines 347-349. Authors should include uninfected TG from mice that had not been explanted (i.e. fixed immediately after removal) to control for the effects of explant on beta-catenin expression. 

Response: These images were included as the uninfected panels shown in Figure 1 of the original manuscript. 

Concern #5: Sentence on lines 279-280. What are the numbers of latently infected TG neurons in animals infected with WT virus compared to LAT mutant virus? Need to show this important control. 

Response: This topic was discussed in the revised manuscript (lines 193-206). 

Concern #6: Figure 6. What is the degree of reactivation from TG explanted from WT and LAT mutant infected animals? Presumably, the medium associated with the explanted TG analyzed for beta-catenin expression shown in figure 6 was available for virus titration. 

Response: Studies in Figure 7 and 8 show a comparison between efficiency of wt HSV-1 versus the LAT-/- mutant. The results demonstrated at least 2 logs reduction in the LAT-/- mutant relative to wt. These results are consistent with our recent studies, reference #31 in the revised manuscript.

Concern #7: Figure 7. How do Wnt/beta-catenin inhibitors impact the reactivation of LAT mutants? 

Response: dLAT2903 explant-induced reactivation is significantly reduced compared to WT (reference 31 and Figures 7 and 8). Hence iCRT14 was assessed for LAT mutant reactivation at 5 days after explant (the peak effect of iCRT14 observed in WT HSV1 reactivation). dLAT2903 reactivation was decreased in the presence of iCRT14: Figures 7 and 8 (lines 349-369). 

Concern #8: How would DEX impact the action of the Wnt/beta-catenin inhibitors on virus reactivation? 

Response: New studies showing this data are presented in Figure 8.

Concern #9: Lines 341-343. This is an egregious over-interpretation of the data.

 Response: Agreed. Our thought on this topic have been revised and are less speculative (lines 393-410). 

Minor issues:

Concern #9: It is curious that Wnt/beta-catenin antagonists are so potent at inhibiting reactivation when the levels of beta-catenin are so low during the reactivation phase. Can the authors comment on this apparent inconsistency? 

Response: The antagonists are measuring viral shedding in the supernatant 4-9 days post-explant. Under the conditions of our TG explant studies, this is when virus shedding consistently occurred (reference 31). Conversely, IHC analyzing beta-catenin expression was performed 4 and 8 hours post-explant. The integrity of the tissue after 16hrs post-explant is compromised and IHC results would not be a clear-cut. Two previous studies demonstrated �-catenin promotes HSV1 productive infection in cell culture (references 32 and 58). This was discussed in the revised manuscript (lines 430-451).

Concern #10: Line 27. Why highlight ocular disease here? Line 22 introduces ocular, nasal and oral infection. Consider "recurrent disease". 

Response: This was modified (lines 29-30).

Concern #11: Line 35. "increased" compared to what? 

Response: This error was corrected in the manuscript (lines 34-36).

Concern #12: Line 47. Odd phrase. Suggests oral disease upon reactivation after primary infection of the eye? This is confusing. Similar issues in abstract. Consider rephrasing. 

Response: This was modified in the manuscript (lines 48-50).

Concern #13: Lines 49-50. This sentence implies LAT is not expressed during lytic infection. I don't think this is the point the authors are trying to make. Consider rephrasing. 

Response: This has been corrected in the manuscript (lines 52-54).

Concern #14: Line 53. Why does the nature of LAT locus products "suggest" functions related to latency? Consider rephrasing.

 Response: This sentence was modified (lines 55-57).

Concern #15: Line 76. “Wnt binding” to what? 

Response: This has been corrected in the manuscript (lines 79-83).

Concern #16: Line 170. “rational” should be “rationale”. 

Response: This has been corrected in the manuscript (lines 172-174).

Concern #17: Line 177. “marker rescued” should read “repaired”. 

Response: This has been corrected in the manuscript (lines 180-182).

Concern #18: Sentence on lines 308-309. Alternatively, the data may indicate that the inhibitors were toxic to the explants. Can the authors control for this? Is there a dose response to the inhibitors? 

Response: Previous data from our lab has shown iCRT14 does not have any cytotoxicity and no significant dose response in cell culture (31). We repeated this study and performed toxicity studies using KYA1797K in cell culture and found this does not have toxic effects in cell culture up to 50µM. This has been updated in the manuscript (lines 328-330).

---

## [Editor Report · Decision Letter 1]

11 Mar 2020

Herpes simplex virus 1 regulates beta-catenin expression in TG neurons during the latency-reactivation cycle.

PONE-D-19-33585R1

Dear Dr. Jones,

We are pleased to inform you that your manuscript has been judged scientifically suitable for publication and will be formally accepted for publication once it complies with all outstanding technical requirements.

With kind regards,

Neal A. DeLuca, Ph.D.

Academic Editor

PLOS ONE
---

## [Editor Report · Acceptance letter]

16 Mar 2020

PONE-D-19-33585R1 

Herpes simplex virus 1 regulates beta-catenin expression in TG neurons during the latency-reactivation cycle. 

Dear Dr. Jones:

I am pleased to inform you that your manuscript has been deemed suitable for publication in PLOS ONE. Congratulations! Your manuscript is now with our production department. 

With kind regards,

on behalf of

Dr. Neal A. DeLuca 

Academic Editor

PLOS ONE